# Chemical reactivity of RNA and its modifications with hydrazine
Nur Yeşiltaç-Tosun [1], Yuyang Qi[1], Chengkang Li[1], Helena Stafflinger [2], Katja Hollnagel [2], Leona Rusling [3], Jens Wöhnert[2], Steffen Kaiser[3] & Stefanie Kaiser [1] ✉

RNA modifications are essential for the regulation of cellular processes and have a key role in diseases such as cancer and neurological disorders. A major challenge in the analysis of RNA modification is the differentiation between isomers, including methylated nucleosides as well as uridine and pseudouridine. A solution is their differential chemical reactivity which enables isomer discrimination by mass spectrometry (MS) or sequencing. In this study, we systematically determine the chemical reactivity of hydrazine with RNA and its native modifications in an aniline-free environment. We optimize the conditions to achieve nearly full conversion of all uridines while avoiding RNA cleavage. We apply the conditions to native tRNA$^{Phe}$ which allows discrimination of pseudouridine and uridine by MALDI-MS. Furthermore, we determine the identity of the reaction product of hydrazine with various modified nucleosides using high resolution mass spectrometry and quantify the reaction yield in native tRNA from *E. coli* and human cells under various hydrazine conditions. Most modified nucleosides react quantitatively at lower hydrazine concentration while uridines do not decompose under these conditions. Thus, this study paves the way to exploit aniline-free hydrazine reactions in the detection of RNA modifications through MS and potentially even long-read RNA sequencing.

RNA is a versatile biomolecule involved in all life processes including gene regulation, translation, information transfer and stress response. The versatility is achieved by the chemical variety contained in cellular RNAs, which is composed of cytidine (C), uridine (U), guanosine (G), adenosine (A) and more than 170 chemical derivatives of the canonical building blocks[1]. Pseudouridine (Ψ) is the most prevalent RNA modification, resulting from the isomerization of uridine catalyzed by pseudouridine synthases (PUS). Ψ is found in all major RNA species (tRNA, rRNA and mRNA) and it enhances RNA stability which influences RNA structure and thus protein interactions[2]. Other modifications include dihydrouridine (D) or simple methylations such as 7-methylguanosine (m$^7$G) or 3-methylcytidine (m$^3$C). Their locations and stoichiometric abundance at target sites are similarly well regulated and their absence (or overabundance) is causative for disease, called modopathies[3].

Due to their immense importance in human health, the detection of Ψ and other modifications has been subject to intensive research. Ψ is an isomer of U and cannot be detected directly by next-generation sequencing (NGS) technologies since both Ψ and U pair successfully with A in reverse transcription and cannot be distinguished. Similarly, the detection within its sequence context by mass spectrometry (MS) is nearly impossible, as Ψ and U have the same mass (Mw 244 g/mol). Similar challenges arise for other modified nucleosides and discrimination of canonical-to-modified is equally challenging as is the discrimination of modified isomers (e.g. 5-methylcytidine (m$^5$C) and m$^3$C). These challenges have led to the development of chemical tools, which react differentially with the isomers[4–7] and are now successfully used for both sequencing[8–13] and MS detection of Ψ[14–16]. Regarding sequencing, some of these chemistries lead to loss of the aromatic base and subsequently, the RNA can be cleaved by aniline which allows the detection of modifications at the cleavage sites[11–13]. Chemistries that do not induce strand-breaks are additionally suitable for mapping modifications using long-read sequencing methods as applied for the detection of Ψ by Oxford Nanopore technology sequencing[17–19].

An interesting approach for discrimination of Ψ from U by NGS was presented in 2020 by Marchand et al.[11]. The discrimination of Ψ and U is based on the reaction of U with hydrazine, which leads to the destruction of U, while Ψ stays intact[4]. Briefly summarized, hydrazine reacts in a nucleophilic addition reaction with the C6 of U (and to a lower extent C), which results in a decomposition of the nucleobase (Fig. 1a). In a second step, aniline is added to the reaction which promotes RNA cleavage at the abasic site[20,21]. Similarly, hydrazine can be used at lower concentrations and at

[1]Institute of Pharmaceutical Chemistry, Goethe-University Frankfurt, Frankfurt/M., Germany. [2]Institute for Molecular Biosciences and Center for Biomolecular Magnetic Resonance (BMRZ), Goethe-University Frankfurt, Frankfurt/M., Germany. [3]Mass Spectrometry Service Unit, Goethe-University Frankfurt, Frankfurt/M., Germany. ✉e-mail: stefanie.kaiser@pharmchem.uni-frankfurt.de

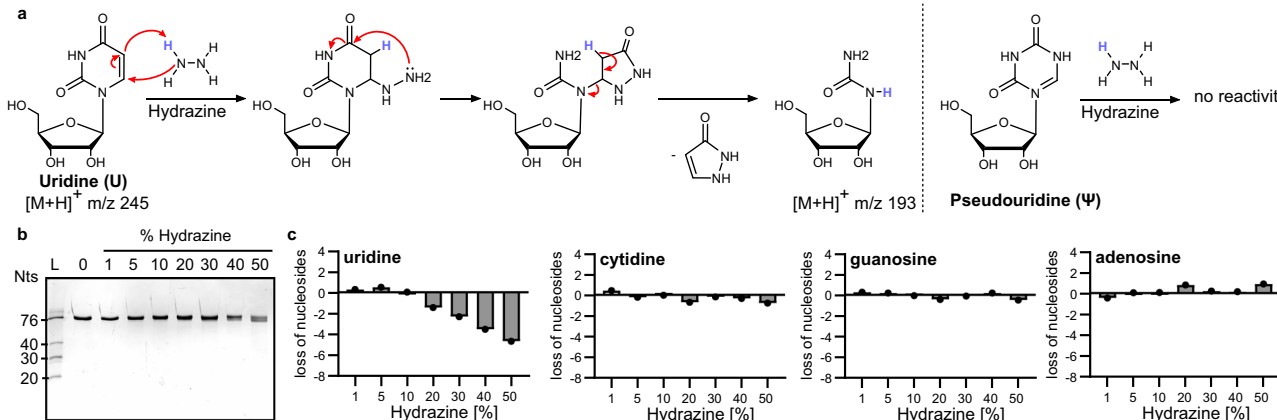

**Fig. 1 | Reaction of uridine with hydrazine in aniline-free conditions.**
**a** Nucleophilic reaction mechanism of hydrazine with uridine. Hydrazine attacks the C6 of uridine which leads to the cleavage of a pyrazolone molecule. The remaining urea–ribose moiety is 52 Da lighter than uridine. Pseudouridine has no sufficient electrophilicity at C6 and does not react with hydrazine. **b** In-vitro transcribed isoleucyl-tRNA from *E. coli* incubated with 1%–50% hydrazine for 1 h at 4 °C (20% TBE urea PAGE, stained with GelRed). **c** Abundance of canonical nucleosides in tRNA$^{Ile}$ after hydrazine treatment quantified by LC–MS/MS analysis. (C cytidine, G guanosine, U uridine and A adenosine).

higher salt strength which causes decomposition of m$^3$C, followed by aniline cleavage and allows its discrimination from m$^5$C by sequencing[13,22].

The success of Ψ and m$^3$C detection by NGS depends on various factors including U-conversion stoichiometry, unwanted RNA cleavage and side reactions with other natural RNA modifications. All these factors are currently unknown and, e.g., the conversion rate of U to its product is unknown, although this conversion rate problematically impacts false-positive calls. Furthermore, the use of hydrazine is currently limited to NGS approaches and detection of Ψ and m$^3$C is not possible by nanopore sequencing thus limiting the platform capabilities.

In this manuscript, we use hydrazine in an aniline-free environment, and we find that RNA cleavage can be fully avoided by performing the reaction at low temperatures. Furthermore, we find that higher U conversion is achieved in dependence of hydrazine concentration, temperature and time. In total, a set of two conditions is identified which provides U conversion of up to 90% while the RNA stays intact. As a proof-of-concept, we show that U and Ψ can be distinguished by MALDI-MS after hydrazine reaction using native tRNA$^{Phe}$ from *Saccharomyces cerevisiae* (*S. cerevisiae*). Concerning hydrazine reactivity, we find that several modified uridines but also cytidine modifications react with hydrazine and are lost from *Escherichia coli* (*E. coli*) or human tRNAs after hydrazine treatment. We use high-resolution MS and determine the reaction products and further the reaction yield using various aniline-free hydrazine conditions. In summary, we show that aniline-free hydrazine chemistry might present a valuable tool for the detection of s$^4$U, acp$^3$U, ac$^4$C, D and mcm$^5$s$^2$U by all currently applied sequencing technologies and MS.

## Results

### Proof of principle: RNA integrity and uridine conversion

Starting from the reports that hydrazine converts uridine to urea-derivatives, we were wondering how the loss of the aromatic nucleobase impacts the stability of the phosphodiester bond in the absence of aniline. We incubated a synthetic 20-nucleotide (nts) long RNA with 50% hydrazine at 4 °C[11] for varying periods of time (1–24 h). Cleavage of RNA, which we define as RNA degradation, can be visualized by classical polyacrylamide gel electrophoresis (PAGE). PAGE of the aniline-free, hydrazine-treated 20-mer showed the full-length 20-mer and no smaller fragments were detected even after 24 h of incubation, indicating that prolonged exposure to hydrazine does not compromise the integrity of the RNA (Fig. S1).

In further experiments, an unmodified transcript of tRNA isoleucine (tRNA$^{Ile}$) was exposed to varying concentrations of hydrazine for 1 h on ice. This 76 nts long RNA did not fragment under hydrazine treatment (Fig. 1b).

After precipitation for removal of excess hydrazine, the RNA was subjected to isotope dilution mass spectrometry (LC–MS/MS)[23] and the number of remaining canonical nucleosides was assessed. Figure 1c shows that neither C, A nor G are lost—not even at 50% hydrazine. U is also not converted in the presence of 1%–10% hydrazine. Starting at 20% hydrazine 1 U (out of 16 U per tRNA$^{Ile}$ molecule) is lost and with rising hydrazine concentrations even more U loss is observed. At 50% hydrazine a maximum of 4 out of 16 Us are lost from tRNA$^{Ile}$. Thus, in our hands, we find a 25% conversion rate for the conditions commonly used in HydraPsiSeq[11]. Another construct, which was in vitro pseudouridylated by TruB at position 55 (tRNA$^{Ile}$ ΔD, AC-arm), was used as a first modified RNA model. Quantification of canonical nucleosides and pseudouridine following hydrazine treatment confirmed that Us are lost while pseudouridine (and other nucleosides) were not compromised by the treatment (Fig. S2). Again, the RNA stayed intact and no shorter RNA fragments were observed. In conclusion, our findings confirm that aniline-free hydrazine selectively degrades uridine without affecting other nucleosides or compromising RNA integrity.

### Optimization of reaction conditions for complete uridine conversion

Using the HydraPsiSeq conditions, we could show that on average 4 out of 16 Us in in-vitro transcribed tRNA$^{Ile}$ undergo conversion while pseudouridine stays intact. A higher conversion of U would enhance the U-Ψ discrimination potential and reduce false-positive Ψ assignments. To achieve a full conversion of every U, we systematically tested variations of the reaction conditions. We examined the effects of incubation time and temperature using 30% hydrazine as a starting point. Figure 2a illustrates that extending the incubation time from 1 to 6 h on ice enhances uridine loss as determined by LC–MS/MS analysis while other canonical nucleosides do not react (Fig. S3A). RNA integrity was maintained, as confirmed by gel electrophoresis in Fig. 2b. It is noteworthy that we see a faster migrating band with longer incubation times which correlates with the substantial loss of Us from the RNA. This shift is most likely caused by the loss of pyrazolone from U which reduces the molecular weight of the Us in RNA by 12%. As we see no sign of shorter fragments or a smear on the gel, this shift is not considered RNA degradation. Next, we tested the impact of temperature on the reaction. Figure 2c, d demonstrates that higher incubation temperatures, e.g. 50 °C, lead to nearly complete loss of all Us (15/16) after both 1 and 6 h of incubation. However, at 50 °C cytidine is lost as well and in addition the quantification becomes obscure as suddenly more G can be detected (Fig. S3B). The explanation for the obscure quantification results can be found in the respective gels in Fig. 2c, d. Here it becomes clear that the RNA

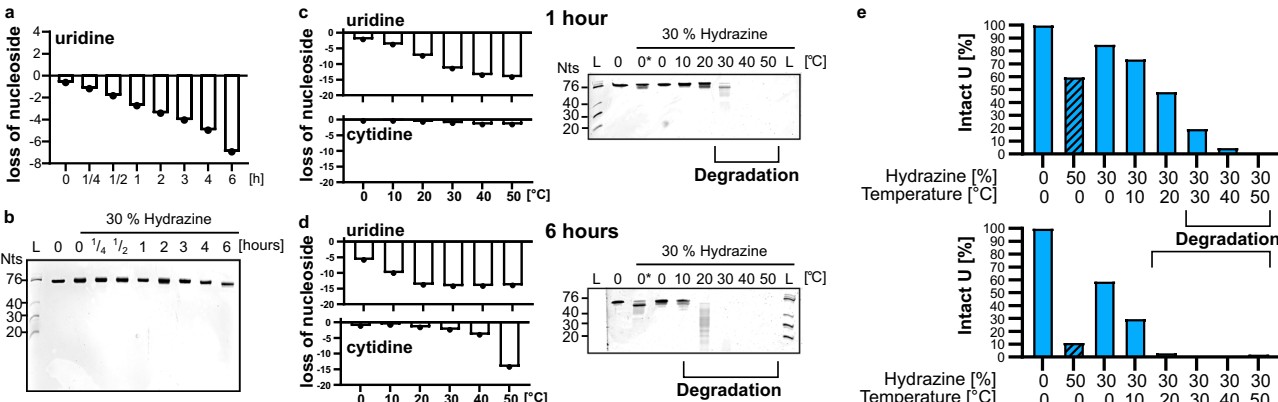

**Fig. 2 | Impact of incubation time and temperature on uridine conversion and RNA integrity. a** Abundance of uridine in tRNA[Ile] after 30% hydrazine incubation for 15 min up to 6 h. **b** 20% TBE–urea polyacrylamide gel, stained with GelRed, of tRNA[Ile] after 30% hydrazine incubation for 15 min up to 6 h. **c, d** Temperature dependence: abundance of uridine and cytidine in tRNA[Ile] after 30% (0* = 50%) hydrazine incubation for 1 h (**c**) or 6 h (**d**). 20% TBE–urea polyacrylamide gel, stained with GelRed, of tRNA[Ile] after 30% (0* = 50%) hydrazine incubation for 1 h (top) or 6 h (bottom). **e** Summary of the optimization experiments indicating the remaining abundance of U on the y-axes in the tRNA after 1 h (top) and 6 h (bottom).

is fully degraded at high incubation temperatures, as shown by the smear of faster migrating RNA fragments on the gel. From these experiments, we can see that RNA remains stable at 0 and 10 °C at 30% hydrazine concentration after both 1 and 6 h of reaction. Incubation with 30% hydrazine for 1 h at 20 °C does not lead to pronounced degradation while 30 °C does. After 6 h of incubation at 20 °C the RNA is clearly fragmenting at this temperature as well. These results indicate that an improvement of U conversion at 0 and 10 °C upon longer incubation times improves U conversion at 0 and 10 °C, while higher temperatures lead to RNA degradation, highlighting the need to balance the conditions for optimal results.

In a final step, we combined the results from the optimization experiments which results in Fig. 2e. Here, we plotted the remaining abundance of U in tRNA[Ile] under the various conditions. Approximately 40% uridine conversion can be achieved within 1 h at either 50% hydrazine/0 °C or 30% hydrazine/20 °C, with the latter condition yielding even higher conversion rates of more than 50% without causing RNA degradation. By extending the incubation time to 6 h, more than 70% conversion occurs at 30% hydrazine/10 °C, while even more than 90% conversion is achieved at 50% hydrazine/0 °C without degrading the RNA.

## Aniline-free hydrazine treatment allows analysis of RNA by MALDI-MS

Next, we wanted to assess whether aniline-free hydrazine treatment is suitable for oligonucleotide mass spectrometry (ONTS-MS). For this purpose, we incubated a synthetic 20-nts long RNA with 30% hydrazine at 0 °C for 1 h. The RNA was precipitated and analyzed by MALDI-MS analysis. The control showed the expected singly charged oligonucleotide at $m/z$ 6550 and to a minor degree the doubly charged species at $m/z$ 3276 (Fig. S4). After hydrazine treatment, three additional peaks arise which differ in mass by 52 Da (Fig. S5). Additional peaks that would indicate unwanted RNA fragmentation are not observed in either MALDI spectrum. The mass difference of 52 indicates the conversion of uridine into urea–ribose (Fig. 1a). In total, we observe the loss of 1, 2 and 3 Us, with the peak corresponding to the loss of 1 U being the main product. To address the question how RNA structure might protect certain Us from conversion, we incubated an unmodified tRNA with hydrazine, performed an RNase T1 digestion and compared the MS results to the corresponding non-hydrazine treated but T1 digested tRNA. In Fig. 3a, the peaks of two U-containing ONTS are shown before and after hydrazine treatment. The first one is the ONT from U61-G69 and contains 2 Us. After hydrazine treatment two new peaks arise which are 52 or 104 Da lighter due to U conversion compared to the untreated tRNA. The second ONT ranges from C28-G35 and contains 1 U. Here, only one new peak with a $\Delta m/z$ of $-52$ compared to the untreated

tRNA is observed which again, indicates U conversion. In total, we could identify six ONTS from the tRNA hydrolysate covering all parts of the tRNA (Fig. S6). This indicates that RNA structure is not a factor in the conversion of U, as we observed products in all U-containing ONTS. Yet, unmodified tRNA has a less stable 3D structure compared to native, modified tRNA. Therefore, we incubated commercially available *S. cerevisiae* tRNA[Phe] with hydrazine and performed RNase T1 treatment. Here, we can identify two ONTS, namely A[Cm]U[Gm]AA[Y]AΨ[m⁵C]UG and [m¹A]UCCACAG which is in accordance with the literature[15]. The first ONT contains 2 Us and 1 Ψ. As seen in Fig. 3b, we see a conversion of 2 Us (Δ−52 and Δ−104) and no third conversion which indicates that U39 is fully pseudouridylated. Interestingly, we also see that Y, the hypermodified nucleoside wybutosine, shows no reactivity toward hydrazine. The second ONT contains 1 U and indeed we see conversion of U by the occurrence of a peak with Δ−52. In conclusion, we observe U conversion in both tRNAs in loop regions but also stem regions which indicates that U conversion can also occur in highly structured RNAs.

While MALDI-MS is a highly useful technology to analyze purified RNAs quickly, it does not allow for automated spectrum interpretation like NASE[24]. To test the compatibility of aniline-free hydrazine treatment with NASE, we treated a synthetic 20-mer with 20% hydrazine for 1 h at 0 °C. The product was separated on a nano-flow reverse phase chromatographic system before injection for high-resolution mass spectrometry (HRMS). The resulting HRMS peaks are shown in Fig. S7. The unreacted RNA showed three clear peaks in the MS which correspond to the expected precursor ion of the full-length RNA in a +4 charge state (charging with either four protons, $m/z$ 1614.23, or three protons +1 of either Na⁺ or K⁺). After hydrazine treatment, the unreacted 20-mer is still clearly detectable at $m/z$ 1614.23. Additional peaks are now present in the spectrum at $m/z$ 1601.22 and at lower abundance at $m/z$ 1588.22. The difference among the peaks at the +4 charge state is 13.01 and 26.01, respectively. This corresponds again to a mass loss of 52 Da and urea–ribose formation. We selected the precursor ion of $m/z$ 1588.22 for higher-energy collisional dissociation (HCD) fragmentation. The fragment spectra were successfully analyzed using NASE[24] and the locations of the converted Us were assigned, e.g. to positions 5 and 8 (Fig. S7). In total, MS/MS analysis hinted toward a random distribution of U conversion.

## Impact of hydrazine treatment on RNA modifications in native RNA

In Fig. 3b, we observed that modified nucleosides, such as Y, 2′-O-methyl-cytidine (Cm), 2′-O-methylguanosine (Gm), m⁵C, 1-methyladenosine (m¹A) and Ψ do not react with hydrazine under aniline-free conditions. We next assessed how hydrazine treatment impacts up to 25 native modifications by

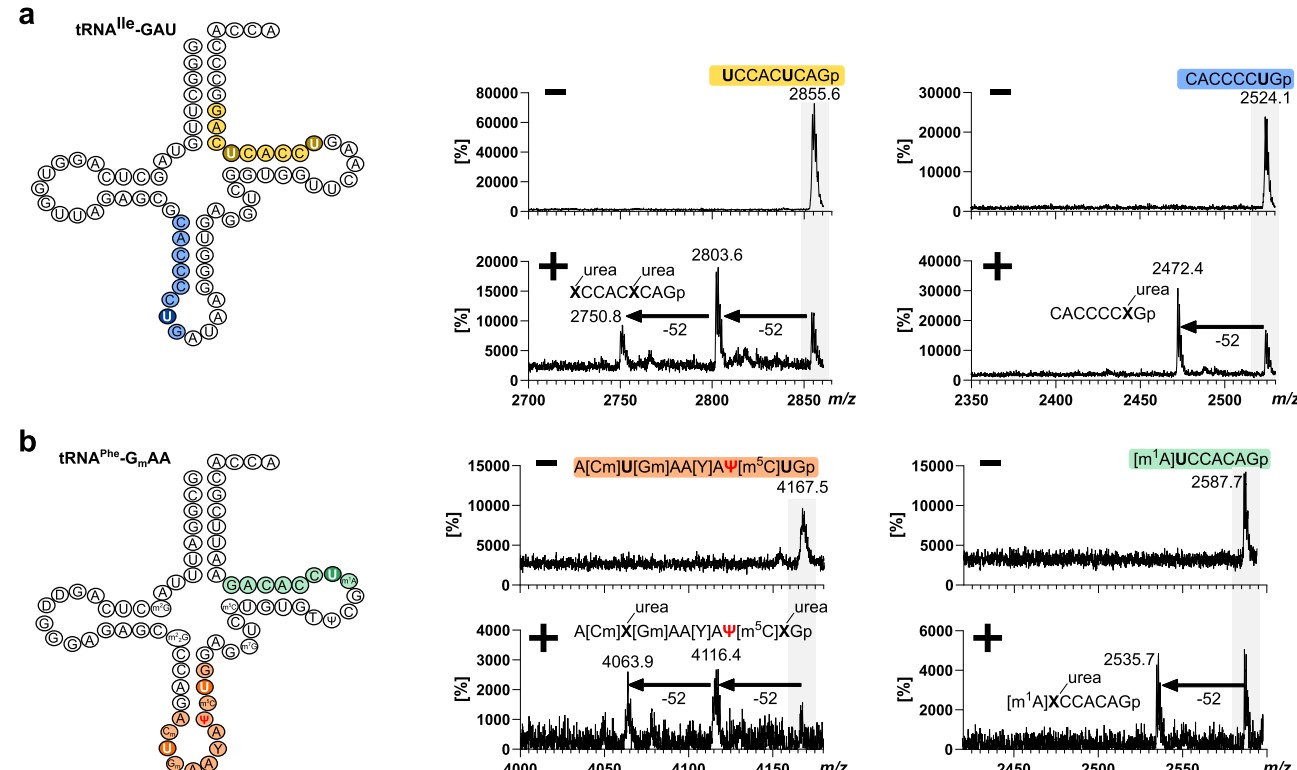

**Fig. 3 | MALDI-MS analysis of hydrazine-treated tRNA. a** Cloverleaf display of unmodified tRNA[Ile] and MALDI-MS spectra of untreated (−) and hydrazine-treated (+) RNase-T1 hydrolyzed tRNA. Full spectra with annotation of all peaks can be found in Fig. S6. **b** Cloverleaf display of native S.c. tRNA[Phe] and MALDI-MS spectra of untreated (−) and hydrazine treated (+) RNase-T1 hydrolyzed tRNA. Hydrazine reaction conditions were 30% hydrazine, 0 °C, 1 h.

incubating native total tRNA from either *E. coli* or HEK cells with hydrazine. As we expect a higher reactivity of modified nucleosides with hydrazine, we chose a low concentration of hydrazine (30%) and incubated for only 1 h at 0 °C. We used our established LC–MS/MS workflow[23] and quantified the abundance of 20 modified nucleosides from *E. coli* tRNA and 23 modified nucleosides from human tRNA. For normalization of the amount of injected RNA, we used the signal of guanosine. As expected, pseudouridine is unaltered by the incubation with hydrazine (Fig. 4a, b) as is 5-methyluridine (m5U). Ribose-methylated uridine (Um) reacts partly with hydrazine. In addition, thiolated uridine modifications such as 4-thiouridine (s4U) in *E. coli* and 5-methoxycarbonylmethyl-2-thiouridine (mcm5s2U) are lost after hydrazine treatment. In contrast, 2-thiocytidine (s2C) from *E. coli* does not react under the 30% hydrazine conditions. In both, *E. coli* and HEK, dihydrouridine (D) is decreased at least tenfold and N4-acetylcytidine (ac4C) is completely lost. Similarly, 3-methylcytidine (m3C) is no longer detectable in human tRNA after hydrazine treatment. With regard to m7G and 6-threonylcarbamoyladenosine (t6A), we find a differential behavior in dependence of the organism. While m7G decreases fourfold in human tRNA (*p* value < 0.0001), no statistically significant loss is observed in tRNA from *E. coli*. For t6A, we find a loss in *E. coli* tRNA, but not in human tRNA. Chemically, there is no difference between *E. coli* m7G and human m7G, or t6A, respectively, and thus the result does not reflect differential hydrazine selectivity but potentially other factors such as accessibility of the nucleosides[20,25] or the presence of catalysts that enhance the reaction.

Intrigued by the observation that some modifications are fully lost, while others are only partially lost, we repeated the experiment with *E. coli* and HEK total tRNA with different hydrazine concentrations at 0 °C for 1 h. In accordance with Fig. 2e, we see a 50% conversion of U under the highest hydrazine concentration and no U conversion at concentrations below 10% (Fig. S8). Here, only one replicate was analyzed due to the sigmoidal decrease of modifications with rising hydrazine concentrations. The hydrazine-sensitive modifications are more reactive than U as their abundance decreases even at low hydrazine concentrations (Fig. 4c). For example, at 10% hydrazine, s4U, m3C and mcm5s2U are nearly fully lost, while only 5% of U is converted. These modifications appear to have the highest reactivity with hydrazine. At 30% hydrazine, ac4C and D are fully lost, m7G is reduced to ~25% of its original abundance, and acp3U decreases by ~40%. At even higher concentrations, s2C is also lost from the tRNA. Figure 5 summarizes the differential chemical reactivity of the hydrazine-sensitive nucleosides. Regarding m7G and t6A which appeared to be differentially reactive in regard to the tRNA's organismal origin, we see a comparable reactivity of m7G in *E. coli* and HEK tRNA at all concentrations, except at 30% hydrazine (Fig. S8). It is not yet fully understood why m7G exhibits this reproducible but unexpected behavior at this particular hydrazine concentration in *E. coli* tRNA, while human tRNA shows the expected sigmoidal concentration dependence (Fig. 4 and Fig. S8). Similarly, t6A is still behaving unexpectedly. Again, we see loss of t6A from *E. coli* tRNA, even at the lowest hydrazine concentrations, while t6A does not change in abundance in HEK-derived tRNA. t6A is uniquely found at position 37 in both organisms' tRNAs and therefore the modification is perfectly accessible to hydrazine. While assuming that the chemical structures of t6A are identical in bacterial and human tRNA, we conclude that the bacterial tRNA preparation might contain a cofactor, that catalyzes the hydrazine-dependent decomposition of t6A.

In summary, we see that aniline-free hydrazine treatment enables the reaction of modified nucleosides even at hydrazine concentrations that mainly prevent U conversion. Thus, hydrazine might be a valuable reagent for sequencing-based detection of modified nucleosides.

## Reaction products of modified nucleosides with hydrazine

For the use of hydrazine in MS- or sequencing-based RNA modification analysis, it is important to know the yield (Fig. 4) of the reaction but also the reaction products. For this purpose, we took commercially available modified nucleoside lyophilized reference standards, dissolved them in

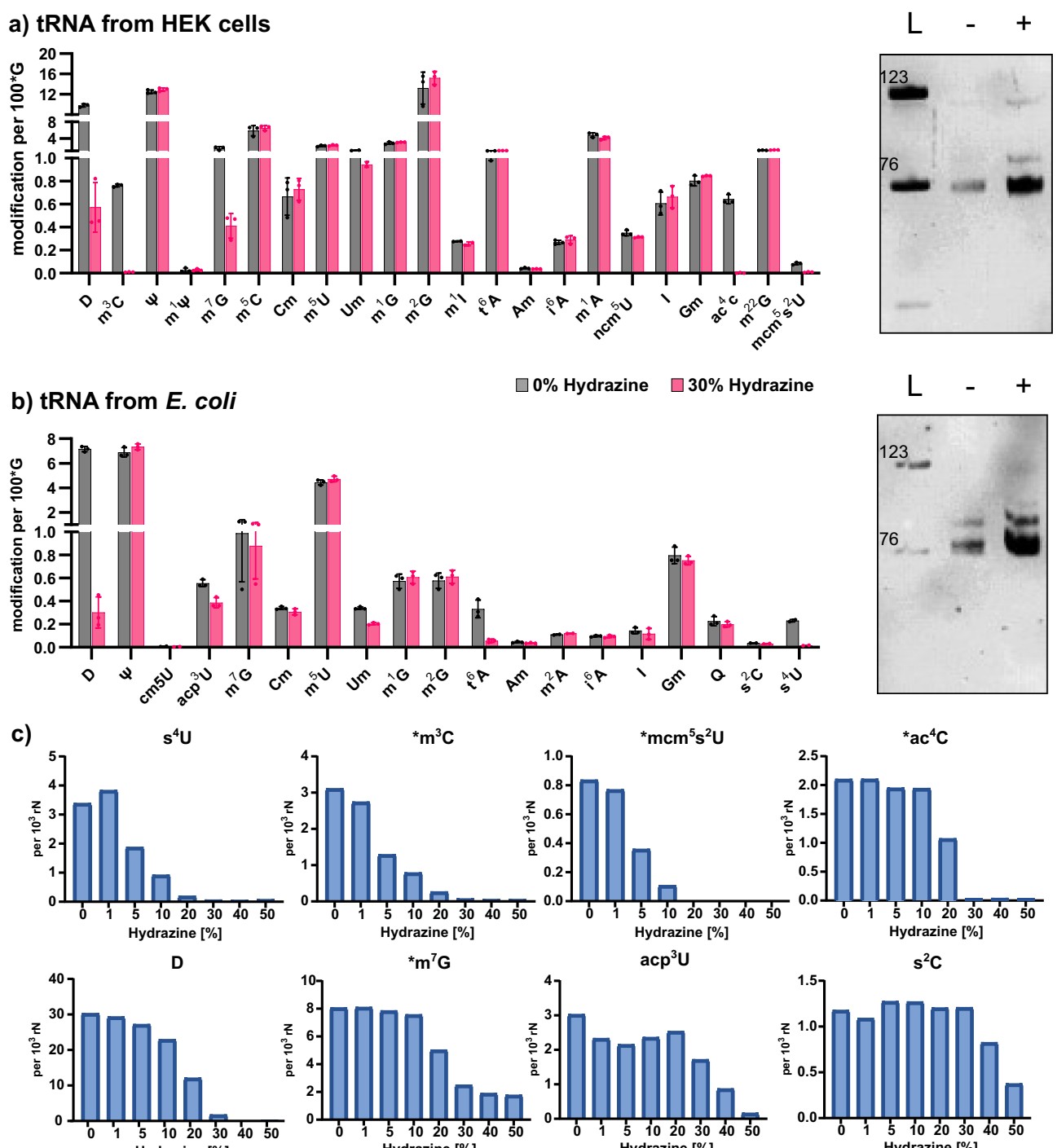

**Fig. 4 | Absolute quantification of modified nucleosides in total tRNA. a** Absolute abundance of modified nucleosides from human embryonic kidney cells (HEK) tRNA (left) and tRNA integrity on a 10% PAGE (right). **b** Absolute abundance of modified nucleosides from *E. coli* tRNA (left) and tRNA integrity on a 10% PAGE. **a, b** Average of *n* = 3 biological replicates, error bars reflect standard deviation.

L = ladder of 76 and 123 nts long RNAs. **c** Abundance of nucleosides after hydrazine treatment (1 h, 0 °C) using rising hydrazine concentrations. Data taken from *E. coli* total tRNA unless marked with *, which indicates data from human total tRNA. From *n* = 1 replicate.

water and performed the aniline-free hydrazine reaction (0 °C, 1 h, 30%). After the removal of hydrazine by freeze-drying, the nucleosides were analyzed by high-accuracy TOF-MS (time-of-flight). Note that mcm⁵s²U and t⁶A could not be used due to their current unavailability as dry powder. Starting with ac⁴C, here we see C as the main product which indicates a hydrazine-induced deacetylation. Similar to U, the C6 appears to react as an electrophile in the case of acp³U, m³C, ac⁴C and s²C which lose the pyrazolone moiety after hydrazine reaction, potentially following the same

reaction mechanism as U (Fig. 1a). The reaction products are shown in Fig. 5, HRMS spectra in Table S1, MS/MS spectra in Table S2 and theoretical mass and mass error in Table S3. The reaction mechanisms of s⁴U, D and m⁷G are unique and shown in Fig. 6a–c. From the MS data, we propose that C4 acts as the electrophile for both D and s⁴U due to the lower electrophilicity of C6 of these modifications[21]. Subsequently, D and s⁴U add hydrazine to C4 of the nucleobase. In the case of s⁴U, the sulfur is subsequently eliminated and thus the aromaticity of the base is maintained

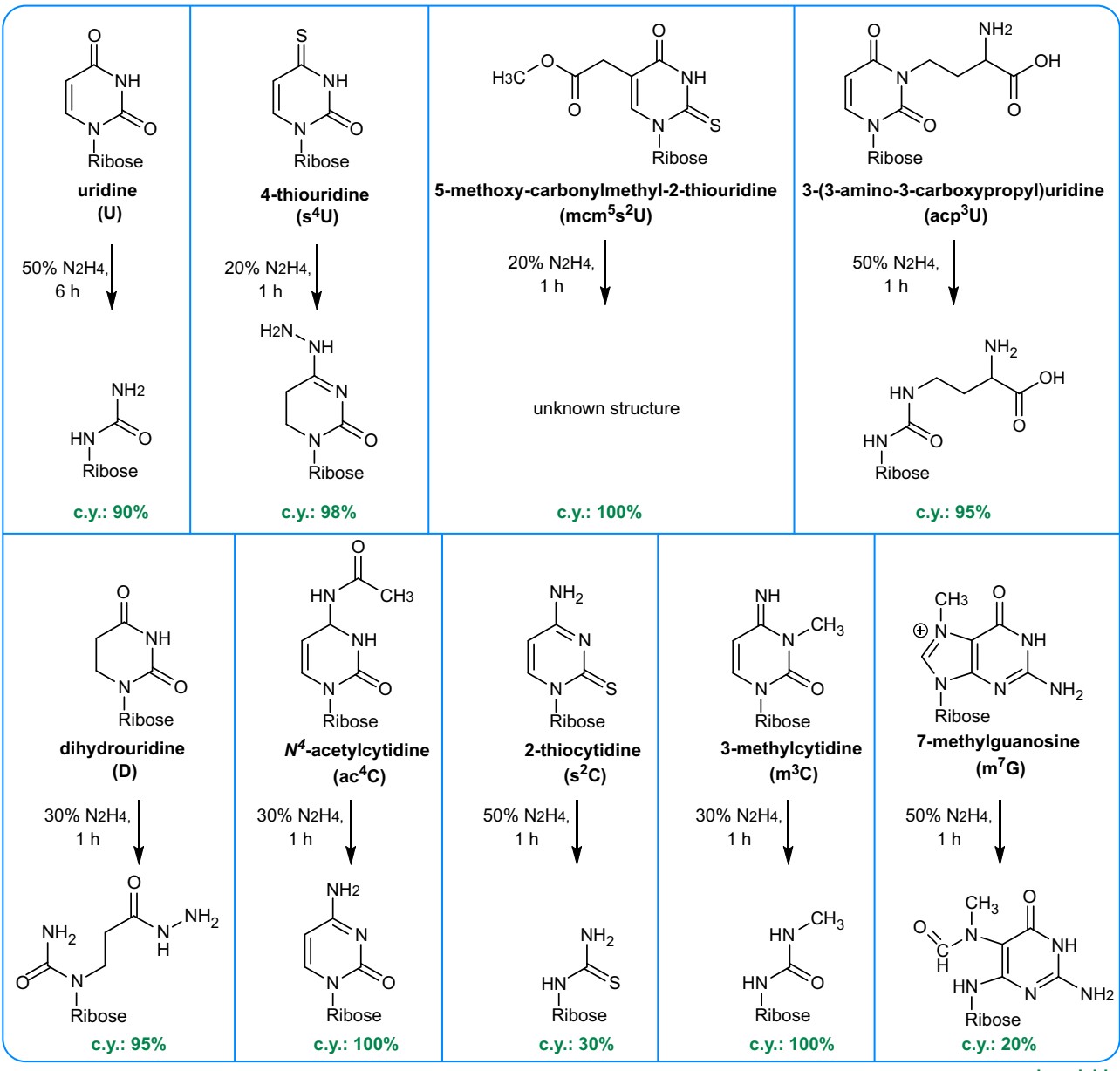

**Fig. 5 | Reaction products and yields for modified nucleosides treated with hydrazine.** Each box depicts the nucleoside and its product after hydrazine conversion. The concentration of hydrazine and incubation time for near-complete conversion is indicated next to the arrow. The conversion yield (c.y.) is given below the product in green font. The reaction product structures of Uridine[4,7], m³C and m⁷G[12] were previously reported.

(Fig. 6a). The nucleoside loses the nucleobase in MS² and further fragmentation of the nucleobase (MS³) reveals loss of ammonia (−17) and hydrazine (−32) which further confirms the structure. As the accuracy is fairly low for this product (15.6 ppm) but the fragmentation spectra appear to fit (4.3 ppm), we are confident in the product's structure. In alkaline environments D is reported to hydrolyze with subsequent ring opening[26]. We assume this is the first step, before hydrazine adds to the C4 which is supported by the accuracy of 2.9 ppm in the HRMS (Fig. 6b). In MS², we observe again the neutral loss of ribose and the remaining construct undergoes loss of ammonia and hydrazine in MS³ which substantiates our proposed structure. Furthermore, the classical HNCO loss (−43) is observed from classical uracil nucleobase fragmentation[27]. Finally, m⁷G does not react with the hydrazine molecule but instead with hydroxyl-ions (or hydroxyl-radicals if bivalent cations are present in the solution[28]). Thus, the well-known reaction product N7-methyl-FAPy-G (Fig. 6c) is formed which is confirmed by our HRMS (MS1 7.5 ppm and MS² 3.8 ppm) and

MS³ spectra[29]. This product is also formed under alkaline treatment as recently reported for AlkAnilinSeq[12].

## Discussion

The detection of Ψ and modified nucleosides through sequencing and MS is a major challenge in the analysis of an organism's epitranscriptome. The discrimination of the isomers Ψ and U is possible by exploiting their differential chemical reactivity, e.g. through alkylation of Ψ[4], bisulfite addition to Ψ[6], 2-bromoacrylamide-assisted cyclization sequencing (BACS)[10] or destruction of U[4]. While the chemical labeling of Ψ was exploited by MS[14,15], NGS[8] and nanopore[17] sequencing, the destruction of U was so far only usable for NGS due to the strand-breaks induced by the aniline cleavage[11]. Similarly, the detection of m³C was also only possible in conjunction with aniline cleavage and NGS[13,22]. Here we show that aniline-free hydrazine chemistry allows the reaction with at least eight modified nucleosides while the RNA phosphodiesters remain intact.

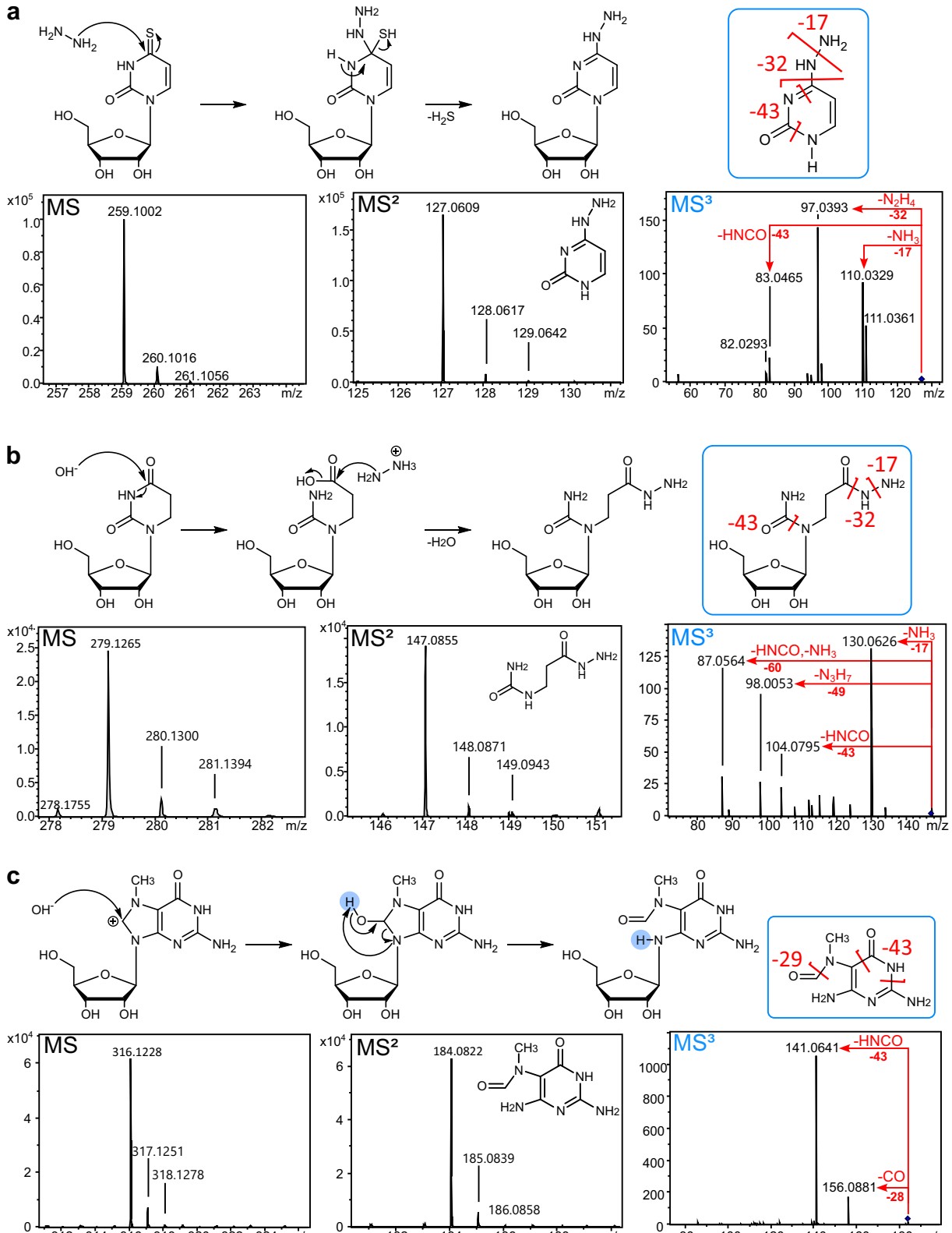

**Fig. 6 | Proposed reaction mechanism of modified nucleosides with hydrazine supported by high-resolution mass spectra and MS²/MS³ fragmentation spectra.** Reaction mechanisms are given above the MS1 spectrum of the intact nucleoside product (left), MS2 spectra of the nucleobase product (middle) and MS3 spectra of the fragmented nucleobases (right). The fragmentation reactions of the nucleobases are indicated in red within the MS3 spectra and in the nucleobase structure given above (blue boxes). **a** 4-thiouridine (s⁴U) **b** dihydrouridine (D) **c** 7-methylguanosine (m⁷G).

We found that changes to both, incubation time and hydrazine concentration, are well tolerated by the phosphodiester and RNA stays intact. In contrast, high temperatures exceeding 10 °C lead to RNA degradation and we recommend a reaction on ice for hydrazine treatment. With our conditions a destruction of up to 90% of all Us in an RNA is possible. While a high conversion rate will reduce the risk of false-positive assignment of Ψ sites by NGS, the near-complete absence of U might disturb the reverse transcription process itself which makes high conversion rates less efficient with low read counts as a potential consequence. For HydraPsiSeq conditions we anticipate a conversion rate of ~25%, which is apparently low enough to still allow reverse transcription. In contrast, nanopore sequencing does not depend on reverse transcription and here, a high U conversion rate would be beneficial for reliable Ψ detection. Yet, judging from recent developments in the field, novel chemistries are now available for Ψ detection which will most likely reduce the use of hydrazine for Ψ detection[10,17–19].

Yet, our data clearly indicate the usefulness of hydrazine to study RNA modifications, as already showcased for $m^3C$ by the Gregory lab[13].

Here, we systematically determine the yield of aniline-free hydrazine reactions with important RNA modifications such as D, $s^4U$, $mcm^5s^2U$, $ac^4C$, $m^7G$[25], $m^3C$[13] and we found additional reactivity with $acp^3U$ which was unreported to the best of our knowledge. Importantly, we find that most modified nucleosides, especially D, $s^4U$, $mcm^5s^2U$, $ac^4C$ and $acp^3U$ react more rapidly than U which is important to maintain reverse transcription activity in NGS approaches. According to our HRMS data, all reaction products are unlikely to engage in regular base pairing which would allow their detection as mismatches in NGS sequencing. Furthermore, we assume that most reaction products will enable their detection by nanopore sequencing as well, since they will have a distinct impact on the current inside the nanopore. Future studies will examine this hypothesis in more detail.

Besides RNA sequencing, MS of RNA is an important tool for the direct detection of RNA modifications within their sequence context. We could show that aniline-free hydrazine chemistry is compatible with both MALDI-MS and nano-LC HRMS of oligonucleotides. Importantly, the combination of high-resolution MS and subsequent analysis with NASE software[24] confirmed that hydrazine-treated RNA is effectively analyzed, with clear identification of uridine modifications as urea–ribose. This indicates that hydrazine treatment is a viable approach for MS-based shotgun RNomics workflows and thus comprehensive RNA modification analysis. Here, we plan to use the aniline-free hydrazine chemistry to distinguish the location of $m^5C$ and $m^3C$ in tRNAs containing both isomers. MS is of particular importance in this respect due to its ability to assign absolute numbers and thus modification yield within the RNA. Our own analyses of purified tRNA isoacceptors by LC–MS analysis confirm the HAC-Seq data[30]. Yet, we also want to emphasize the challenges connected with placing modifications in human cytosolic tRNA. Human cytosolic tRNAs are composed of 47 anticodon families. Out of these 47 families, the modification profile of ~20 is well-known and reported[1]. Yet, the difficulties arise in the detail as each major family has up to 20 isoforms. Thus, approximately 417 different cytosolic tRNAs exist and first reports emerge that isoforms do not only differ in sequence but also in their modification profile[31]. Here, we want to raise attention to a new case found in tRNA Met-CAU, where isoforms 5-1 and 7-1 are clearly found to carry $m^3C$ at position 20[13,22]. We purified the complete tRNA Met-CAU family and quantified the abundance of $m^3C$ alongside other modifications, and we could not detect $m^3C$ in our purified tRNA (Fig. S9). This indicates that the $m^3C$-modified isoforms are of low abundance compared to the C20-unmodified isoforms in this tRNA family. For comparison, we also purified tRNA Leu-CAG, where $m^3C$ is reported at position 47 from the same HAC-Seq studies. Here, we clearly find stoichiometric abundance of $m^3C$ in the purified tRNA family (Fig. S9). Therefore, we want to advocate for a clear denomination of tRNAs by their isoform in both primary data[13] and databases[1].

Overall, we conclude that aniline-free hydrazine chemistry has a high potential for the detection of various modified nucleosides using both sequencing and oligonucleotide MS.

## Materials and methods
### Chemicals and reagents
All chemicals and reagents were purchased from Sigma Aldrich (St. Louis, MO, USA) unless otherwise specified. Nucleoside standards, including pseudouridine (Ψ), 1-methyladenosine ($m^1A$), N3-methylcytidine ($m^3C$), N7-methylguanosine ($m^7G$), $m^5C$, $m^5U$, 2′-O-methylcytidine (Cm), 2′-O-methylguanosine (Gm), 2′-O-methyladenosine (Am), 2′-O-methyluridine (Um), 1-methylguanosine ($m^1G$), 2-methylguanosine ($m^2G$), 2,2-dimethylguanosine ($m^{22}G$), inosine (I), and 5-carbamoylmethyluridine ($ncm^5U$), were sourced from Carbosynth (Newbury, UK). N6-threonylcarbamoyladenosine ($t^6A$) was obtained from TRC (New York, Canada). N6-isopentenyladenosine ($i^6A$) and queuosine (Q) were generously provided by the Dedon lab, and 1-methylinosine ($m^1I$) was a generous gift from STORM Therapeutics Ltd. (Cambridge, UK). The nucleosides 2-thiouridine ($s^2U$) and 4-thiouridine ($s^4U$) were purchased from TRC and the nucleoside 2-thiocytidine ($s^2C$) was from Berry and Associates. The 20-mer RNA oligonucleotide (UGAGG-CAGGAGGUUGAAUAG) was purchased from Dharmacon (Lafayette, CO, USA).

### In vitro transcription of *E. coli* tRNA^Ile
Plasmid *pPK1204* with the correctly sequenced tRNA^Ile insert was transformed into *E. coli* DH5α competent cells and cultured on a large scale to produce sufficient plasmid DNA. The plasmid DNA was isolated using a Qiafilter Plasmid Maxi Kit (Qiagen, Venlo, the Netherlands) and linearized with *HindIII* according to manufacturer protocol.

In vitro transcription of tRNA^Ile was conducted using optimized concentrations of DNA template and magnesium acetate. For large-scale transcription, the process was performed under these optimized conditions for a duration of 4 h. Specifically, the conditions included 40 mM magnesium acetate and 100 ng/µL of DNA template.

RNA products, including HDV ribozymes and uncut RNA, were separated by preparative urea–PAGE. tRNA^Ile was then extracted from the gel, ethanol precipitated, and desalted using a PD10 column (Cytiva, Marlborough, MA). The RNA was concentrated using a SpeedVac (Thermo Fisher Scientific, Waltham, MA) and subsequently stored at −20 °C.

### Hydrazine treatment
1 µg of in-vitro transcribed tRNA^Ile was treated with hydrazine (Sigma Aldrich, St. Louis, MO, USA, 207942-100G) at a final concentration of 0%–50% v/v as indicated in the text. The samples were incubated on ice or at elevated temperatures for 1–24 h, followed by the addition of 2% lithium perchlorate ($LiClO_4$) in acetone, at ten times the sample volume, to induce precipitation. After an 8-min incubation at room temperature, the samples were centrifuged at 8000 rcf for 8 min. The supernatant was discarded, and the RNA pellets were washed with 150 µL of 70% ethanol (EtOH). The samples were then centrifuged at 12,000 rcf at 4 °C for 15 min, and the supernatant was removed. The pellets were air-dried for 20 min and resuspended in 20 µL of water for further analysis or storage at −20 °C.

### RNase T1 digestion of tRNA into oligonucleotides
1 µg of tRNA was incubated with 50 U RNase T1 in an ammonium acetate buffer for 30 min as previously described[32]. Afterwards, the enzyme was removed by using the oligo clean&concentrator kit from Zymo according to the manufacturer protocol (#D4060, Zymo Research, Freiburg, Germany).

### MALDI-TOF mass spectrometry of oligonucleotides
For each sample 0.5 µL of 3-HPA matrix (3-hydroxipicolinic, half-acid saturated in $H_2O$:acetonitrile (1:1; v:v) (Bruker Daltonics, Bremen, Germany), containing 10 mg/mL diammonium hydrogen citrate) are spotted on an AnchorChip (800 µm) target. The matrix solution is allowed to dry at room temperature. Then 0.5 µL of each sample (7.75 µM) are spotted on top of the dried matrix preparation spot. The sample is allowed to dry and the spectra are recorded in reflector and positive ion mode on an ultrafleXtreme MALDI-TOF-TOF mass spectrometer (Bruker Daltonics, Bremen, Germany).

## Liquid chromatography (tandem) mass spectrometry (LC–MS)-based oligonucleotide analysis

The LC–MS-based oligonucleotide analysis was conducted on a UHPLC Ultimate™3000 RSLCnano system (Thermo Fisher Scientific, Germering, Germany) coupled with a Q Exactive Plus hybrid quadrupole-Orbitrap mass spectrometer (Thermo Fisher Scientific, Bremen, Germany). A stainless steel emitter (30 µm ID, ES542, Thermo Scientific, Germering, Germany) was mounted onto the Nanospray Flex ion source (P/N ES071, Thermo Scientific, Germering, Germany). In total, 200 ng of hydrazine treated 20-mer (in 5 µL solution) for each treatment condition were injected via the "pre-concentration" injection mode using 10 mM ammonium acetate (pH 5.5) as mobile phase A (MPA) and ACN as mobile phase B (MPB). Samples were first trapped onto a PepMap™ Neo trap cartridge (5 mm × 0.3 mm, 5 µm, 100 Å, cat# 174500, Thermo Scientific Inc., Karlsruhe, Germany) using 100% MPA for 3 min at a 20 µL/min flow rate. Afterward, the pre-concentrated samples were reverse-eluted onto a Acclaim™ PepMap™ 100 C18 HPLC column (150 mm × 0.075 mm, 2 µm, 100 Å, cat# 164534, Thermo Scientific Inc., Karlsruhe, Germany) using a 0.3 µL/min flow rate with gradient at 30 °C. The gradient started with a linear increase from 5% MPB and reached 50% MPB at 40 min. Then MPB ramped up to 99% in 2 min and lasted for 13 min before a rapid drop to 0% in 1 min. The column was then equilibrated at 0% for 14 min before the next injection. Blank injection was included between every two injections using MilliQ water.

Eluted samples were ionized using 2.5 kV spray voltage under positive ion mode at 275 °C and analyzed using the "Full MS/data-dependent MS2 (dd-MS2)" method. Resolution, AGC target, and maximum injection time for both MS and MS2 were set to be 70,000 (FWHM), 1e6, and 300 ms, respectively. $m/z$ within 250–2500 were recorded. The Top 15 most intense ions were sent to the HCD for MS2 fragmentation using 28 normalized collision energy (NCE) with 10 s dynamic exclusion duration with $m/z$ 1.7 isolation window width.

## NucleicAcidSearchEngine (NASE) analysis

The profile spectra of both MS1 and MS2 from the raw files were converted to centroid data as mzML format files using MSConvert (https://github.com/ProteoWizard/), which were then analyzed using the NASE software (OpenMS ver. 3.0.0-pre-nightly-2023-03-09, https://openms.de/) for database matching[24,33,34]. A database (as a fasta file) was created for the search using the sequence of the 20-mer (i.e. 5′-GUAGUCGUGGCCGA-GUGGUU-3′). The false discover rate cutoffs for both target/decoy database searches were set to be 0.01. Mass tolerances for precursor and fragment ions were set to be 10 ppm and 20 ppm, respectively, while +1 to +10 were set as the potential precursor charge states. Potential cation adducts, including sodium ($Na^+$), potassium ($K^+$), and ammonium ($NH_4^+$), were also included in the search. Precursor isotopes of −1, 0, 1, 2 and 3 were set, while all possible fragment ions types, including a-B, a, b, c, d, w, x, y and z ions, were selected.

## tRNA digestion for nucleoside mass spectrometry

Approximately 100 ng of tRNA samples were enzymatically digested into single nucleosides. The digestion reaction included 0.2 U alkaline phosphatase, 0.02 U phosphodiesterase I (VWR, Radnor, PA, USA), and 0.2 U benzonase in a buffer of 5 mM Tris (pH 8) and 1 mM $MgCl_2$. To prevent nucleoside deamination and oxidation, the reaction mixture contained 0.5 µg tetrahydrouridine (Merck, Darmstadt, Germany), 1 µM butylated hydroxytoluene, and 0.1 µg pentostatin. The mixture was incubated at 37 °C for 2 h. Stable isotope-labeled internal standard (SILIS)[35] was added at 1/10 volume to each filtrate prior to analysis via triple quadrupole mass spectrometry (QQQ-MS).

## QQQ mass spectrometry

Nucleosides were quantified using a 1290 Infinity II UHPLC system coupled with an Agilent G6470A Triple Quadrupole mass spectrometer equipped with electrospray ionization (ESI-MS, Agilent Jetstream). The mass spectrometer operated in positive ion mode under the following optimized conditions: skimmer voltage at 15 V, cell accelerator voltage at 5 V, nitrogen gas temperature at 230 °C with a flow rate of 6 L/min, sheath gas ($N_2$) temperature at 400 °C with a flow rate of 12 L/min, capillary voltage at 2500 V, nozzle voltage at 0 V, and nebulizer pressure at 40 psi. Chromatographic separation was performed using a Synergi Fusion-RP column (Phenomenex®, Torrance, California, USA; Synergi® 2.5 µm Fusion-RP 100 Å, 150 × 2.0 mm) maintained at 35 °C and a flow rate of 0.35 mL/min. The gradient elution started with 100% mobile phase A (5 mM $NH_4OAc$, pH 5.3) for 1 min, then increased to 10% mobile phase B (pure acetonitrile) over 5 min, and to 40% B over 7 min. The column was flushed with 40% B for 1 min, followed by re-equilibration at 100% A for 3 min. For MS measurements, the dynamic multiple reaction monitoring mode was used.

## Calibration for absolute quantification of modified nucleosides

For the quantification of nucleosides, a 1:1 serial dilution was prepared. Canonical nucleosides were calibrated with a maximum concentration of 100 pmol. Pseudouridine and dihydrouridine were calibrated at maximum concentrations of 20 pmol, while other modified nucleosides were calibrated at 5 pmol. A total of 12 calibration points were generated for each set of nucleosides.

## Accurate mass determination

To determine accurate mass and structural elucidation of nucleoside products after hydrazine treatment, samples were measured with an UHPLC Ultimate™3000 (Thermo Fisher Scientific, Germering, Germany) coupled to a MicrOTOF-qII (Bruker, Bremen, Germany) with electrospray ionization (ESI) in positive mode. Mobile Phase A was 95:5 water/acetonitrile + 0.1% formic acid; Mobile Phase B was 98:2 acetonitrile/water + 0.1% formic acid. To prepare the hydrazine reactions, 50 nmol of each nucleoside ($m^7G$, D, $s^4U$, $s^2C$, $ac^4C$, $acp^3U$, $m^3C$) were combined with either 3 µL of hydrazine solution (+2 µl pure water, final concentration of 30% hydrazine) or 5 µL of Ultra-pure water (control) with a final nucleoside concentration of 5 mM. Nucleoside-hydrazine solutions were incubated with for 1 h on ice. Afterwards, the solvent was removed by dry freezing. Dried samples were stored at −80 °C until measurement. The resulting nucleosides were solubilized in 10 µL Ultra-pure water with vortex-mixing, followed by dilution to 200 µM and 100 µM with Lockmass Solution (5 µg/mL each carbamazepine, flunarazine and reserpine in acetonitrile). Dilutions of corresponding reference stocks were also prepared with Lockmass Solution. Samples were infused with a syringe pump at 5 µL/min into a mobile phase flow of 50% Mobile Phase B at 200 µL/min. A full $MS^1$ scan was acquired, followed by MRM of the nucleoside for $MS^2$, and finally, a pseudo-$MS^3$ combining in-source collision-induced dissociation (isCID) and $MS^2$ fragmentation of the nucleobase (Table S4). Data were processed and analyzed in Bruker Data Analysis where an internal calibration correction was applied using the Lockmass solution $[M+H]^+$ values (237.10224, 405.21368 and 609.28066 $m/z$ for carbamazepine, flunarizine and reserpine, respectively). All theoretical masses and $m/z$ values were calculated using ChemCalc[36]. Mass error values were calculated using Excel 2016 (Microsoft) (Table S3).

## Reporting summary

Further information on research design is available in the Nature Portfolio Reporting Summary linked to this article.

## Data availability

The data that support the findings of this study are available from the corresponding author upon request.

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

## Acknowledgements

This work was supported by the Deutsche Forschungsgemeinschaft [325871075-SFB 1309 to Stefanie Kaiser]. We are grateful to Peter Watzinger for providing TruB and to Peter Kötter for providing the pPK1204 plasmid. Additionally, we would like to thank to Dr. Alexandre Vicente and Kira Kerkhoff for providing total tRNA of *E. coli* and HEK cells. This project is funded by the European Regional Development Fund as part of the Union's response to the COVID-19 pandemic (EFRE-React).

## Author contributions

Nur Yeşiltaç-Tosun: conceptualization, experimental analysis, sample preparation, MS-data analysis, writing—review and editing. Yuyang Qi: experimental analysis, sample preparation, and MS-data analysis. Leona Rusling: high accuracy MS. Chengkang Li: nLC-QEx-Orbitrap measurement. Helena Stafflinger: sample preparation. Katja Hollnagel: sample preparation. Jens Wöhnert: project administration, writing and editing. Steffen Kaiser: MALDI-MS measurements. Stefanie Kaiser: conceptualization, project administration; writing—review and editing.

## Funding

## Competing interests

The authors declare no competing interests.
