## [Transparent Peer Review file · Communications Chemistry]

Chemical reactivity of RNA and its modifications with hydrazine

Corresponding Author: Professor Stefanie Kaiser

Version 0:

Reviewer comments:

Reviewer #1

(Remarks to the Author)

Investigating the functions of RNA modifications relies on sensitive detection methods. In this study, Kaiser and colleagues have optimized the reaction conditions for hydrazine treatment, achieving the conversion of uridine to abasic sites while leaving pseudouridine intact. Overall, the concept of this study is interesting, and the manuscript is well-organized. However, several major concerns need to be addressed.

Specifically:

1. The principle of this strategy is similar to bisulfite treatment. However, the data in this study show that the conversion rate, even under optimized conditions, is approximately 50% (Figure 4A). Given that this is an indirect detection strategy—identifying pseudouridine by detecting unconverted uridines—and that the stoichiometry of pseudouridine is around 10%, the limited conversion efficiency may lead to potential false positives, making it challenging to apply this method for pseudouridine detection.
2. Figure 2A shows that RNA integrity after 24 hours of treatment is better than after 8 hours of treatment. The authors should provide an explanation for this unexpected result or include higher-quality, more convincing data.
3. In Figure 3B, significant RNA degradation occurs after 4 and 6 hours of incubation. The description in lines 121-122, which states that "RNA integrity was maintained, as confirmed by gel," is therefore inaccurate and should be revised.
4. In Figure 4, 50% of uridines remain intact. Does this mean that half of the uridines at a single site remain unconverted, or does it imply that some uridines are fully converted while others remain unchanged? Additionally, the authors should clarify whether this reaction exhibits sequence or structural bias.
5. All the data presented in this study were obtained via LC-MS/MS or MALDI-TOF. While mass spectrometry is a powerful tool for identifying modifications, the authors propose that this method could be applied to sequencing-based strategies. To support this claim, they should provide data demonstrating a transcriptome-wide application of their strategy or, at minimum, present locus-specific detection results for tRNA or another species of RNA.

Reviewer #2

(Remarks to the Author)

The authors performed the optimization of hydrazine reaction to reduce RNA degradation and uridine conversion yield, which is great of improvement. The novelty is they found other RNA modification could be reacted with hydrazine, which provides the possibility for the detection application, however, the detailed chemistry is missing. Considering the readers interested in Communication Chemistry, the authors should put more results related with chemistry. In my opinion, they have to put all detailed figures and results for new hydrazine reaction on other RNA modifications, why some modification disappeared, including reaction figure, the conversion yields of other RNA modification treated with hydrazine, mass spectrometry to confirm the new products.

Reviewer #3

(Remarks to the Author)

In this manuscript, the author investigates the chemical reactivity of hydrazine with RNA, focusing particularly on uridine. The study identifies optimized conditions that selectively degrade uridine without affecting pseudouridine, potentially useful for pseudouridine detection. However, it's important to note that hydrazine has previously been employed in the development of a quantitative ψ mapping technique named "HydraPsiSeq" (Nucleic Acids Research, 2020, 48, e1110). Building on prior research, this manuscript evaluates hydrazine's chemical reactivity under various conditions, including different hydrazine concentrations and temperature settings. The findings indicate that hydrazine treatment impacts several RNA modifications in tRNA to varying degrees. However, these results also highlight the low specificity of hydrazine treatment, with disparate reactivity across different RNA modifications, posing significant challenges for RNA modification research. Additionally, this study primarily utilizes Mass Spectrometry for detecting chemical reactivity, a method that, while sensitive and useful for identifying modification changes, offers limited insight into the underlying mechanisms or patterns of these reactions. For instance, the differential chemical reactivity at the C6 position of the pyrimidine ring between uridine and cytosine modifications is relatively straightforward, but explaining the alterations of t6A and m7G post-hydrazine treatment remains challenging. While this manuscript contributes to optimizing reaction conditions, it struggles to provide substantial information for advancing the study of RNA modifications. Consequently, I believe it does not meet the publication standards of this journal.

Version 1:

Reviewer comments:

Reviewer #1

(Remarks to the Author)

The authors have addressed all my concerns. I support the publication in communications chemistry.

We are grateful to all reviewer's and their expert comments which inspired us to identify the reaction products of modified nucleosides with hydrazine. We discuss the reaction mechanisms and apply our optimized aniline-free hydrazine treatment in oligonucleotide MS analysis. A detailed response to all comments is found below using blue font and * as an indication of our reply.

Reviewer #1 (Remarks to the Author):

Investigating the functions of RNA modifications relies on sensitive detection methods. In this study, Kaiser and colleagues have optimized the reaction conditions for hydrazine treatment, achieving the conversion of uridine to abasic sites while leaving pseudouridine intact. Overall, the concept of this study is interesting, and the manuscript is well-organized. However, several major concerns need to be addressed.

Specifically:

1. The principle of this strategy is similar to bisulfite treatment. However, the data in this study show that the conversion rate, even under optimized conditions, is approximately 50% (Figure 4A). Given that this is an indirect detection strategy—identifying pseudouridine by detecting unconverted uridines—and that the stoichiometry of pseudouridine is around 10%, the limited conversion efficiency may lead to potential false positives, making it challenging to apply this method for pseudouridine detection.

* We thank the reviewer for this comment. Indeed, among the various options to detect pseudouridine (Burrows and Fleming 2023, Dai, Zhang et al. 2023, Zhang, Jiang et al. 2023, Xu, Kong et al. 2024), hydrazine is probably not competitive. In this manuscript, we do not aim to showcase the usability of hydrazine in pseudouridine detection by sequencing-based detection methods, as this was previously done by others (Marchand, Pichot et al. 2020). Yet, we noticed that in published methods, the actual conversion yield of uridine was unknown. This in turn leads to unknown false positive rates as noted by the reviewer. In our manuscript, we wanted to clearly define the chemical conditions that lead to the highest uridine conversion rate (quantitatively) which in turn will reduce the false positive rate in future sequencing studies. We apologize for not clearly formulating this goal. We revised the manuscript accordingly.

2. Figure 2A shows that RNA integrity after 24 hours of treatment is better than after 8 hours of treatment. The authors should provide an explanation for this unexpected result or include higher-quality, more convincing data.

* Thank you for this comment. In fact, the RNA degradation we want to avoid is caused by cleavage of the phosphodiester bonds which results in a smear on the gel as highlighted in yellow (e.g. from Figure 3d, see on the next page for your convenience).

What can be seen as well (red arrows) is that the hydrazine RNA migrates faster, giving it the appearance of getting slightly shorter. In fact, this is not the RNA degrading, but proof that U converts and pyrazolone is lost (red arrows). Therefore, the hydrazine-treated RNA migrates slightly faster than the untreated and unreacted RNA which is an orthogonal proof of our MS-based observation of concentration dependent U conversion rates.

After clarifying these points, we wish to come back to the original comment of the reviewer: In Figure 2, a lower intensity band can be seen at the 8 hour timepoint (see gel above in red box), which is due to a lower amount of RNA loaded. We apologize for this technical error. A look at the uncropped gel in Figure S1 clearly shows that the RNA is not degrading as no smear is detectable. We have revised the manuscript text to point out our definition of RNA degradation and how we expect RNA degradation to look like by pointing out an example. We thank the reviewer for the comment and we hope our interpretation is now accessible to the readers.

3. In Figure 3B, significant RNA degradation occurs after 4 and 6 hours of incubation. The description in lines 121-122, which states that "RNA integrity was maintained, as confirmed by gel," is therefore inaccurate and should be revised.

* This comment directly follows up on our observation that we do not see strand cleavage and thus a smear (which is our definition of RNA degradation) but rather a shift of the RNA due to the loss of pyrazolone. Thus, this shift is, in fact, proof of the higher U conversion rate orthogonal to our MS data (nucleoside MS and MALDI MS). We have clarified this observation and our interpretation in the manuscript.

4. In Figure 4, 50% of uridines remain intact. Does this mean that half of the uridines at a single site remain unconverted, or does it imply that some uridines are fully converted while others remain unchanged? Additionally, the authors should clarify whether this reaction exhibits sequence or structural bias.

* This is a great question, which we asked ourselves as well. We think, analysis of a structured RNA, such as tRNA by bottom-up MS would allow an answer. Therefore, we have now analysed *in vitro* transcribed tRNA^{Ile} and native tRNA^{Phe} from yeast after RNase T1 by MALDI-MS. We find U conversion in all detected tRNA fragments. While this is a hint towards the hypothesis that hydrazine acts on all sites that are accessible in a random fashion, we think further

investigations of longer RNAs with defined structures is needed. Yet the question is how much impact knowledge on the conversion locations would have for future studies, as we assume that hydrazine treatment for pseudouridine mapping will be replaced by the positive-signal detection methods (BID-Seq, BACS etc.). From our additional new data on other, more reactive modified nucleosides, hydrazine will be most likely used in the future for detection of modified nucleosides at non-uridine destroying conditions. Therefore, we hope the reviewer is satisfied with our additional data on full-length tRNAs.

5. All the data presented in this study were obtained via LC-MS/MS or MALDI-TOF. While mass spectrometry is a powerful tool for identifying modifications, the authors propose that this method could be applied to sequencing-based strategies. To support this claim, they should provide data demonstrating a transcriptome-wide application of their strategy or, at minimum, present locus-specific detection results for tRNA or another species of RNA.

* Thank you for this comment. Indeed, we have started a collaboration with Eva Novoa for showcasing that omitting aniline leads to RNA that can be sequenced by nanopore technology. Especially the detection of modified nucleosides such as m^3C , m^7G , s^4U and acp^3U using low doses of hydrazine, where no U conversion appears, would have significant impact for nanopore-based transcriptome wide mapping of these modifications. We expect a major publishing unit to appear as a follow-up after researching this in full depths with our partners. In this manuscript, we provide the quantitative and chemical foundation (new data for the revision as suggested by reviewer 2) for future nanopore studies. In fact, the Gregory lab has already demonstrated the use of hydrazine for detection of m^3C (Cui, Sendinc et al. 2024) which showcases the importance of knowing the reaction partners of hydrazine in RNA, the yield and products..We hope the reviewer can agree with us, that the readers of Communications Chemistry are most interested in the products of hydrazine/RNA modification reaction which is now the main focus of our work.

Reviewer #2 (Remarks to the Author):

The authors performed the optimization of hydrazine reaction to reduce RNA degradation and uridine conversion yield, which is great of improvement. The novelty is they found other RNA modification could be reacted with hydrazine, which provides the possibility for the detection application, however, the detailed chemistry is missing. Considering the readers interested in Communication Chemistry, the authors should put more results related with chemistry.

In my opinion, they have to put all detailed figures and results for new hydrazine reaction on other RNA modifications, why some modification disappeared, including reaction figure, the conversion yields of other RNA modification treated with hydrazine, mass spectrometry to confirm the new products.

* We thank the reviewer for this important comment that has substantially improved our manuscript and given it significant depths and strengths. We have done all requested experiments, and our manuscript now contains a detailed description of the reaction products (confirmed by HRMS and MS2/MS3 fragmentation) and a hydrazine-dose dependence quantification.

Reviewer #3 (Remarks to the Author):

In this manuscript, the author investigates the chemical reactivity of hydrazine with RNA, focusing particularly on uridine. The study identifies optimized conditions that selectively degrade uridine without affecting pseudouridine, potentially useful for pseudouridine detection. However, it's important to note that hydrazine has previously been employed in the development of a quantitative ψ mapping technique named "HydraPsiSeq" (Nucleic Acids Research, 2020, 48, e110). Building on prior research, this manuscript evaluates hydrazine's chemical reactivity under various conditions, including different hydrazine concentrations and temperature settings.

* Indeed. We want to point out we provide for the first time a quantitative number of converted uridines using different conditions and we provide an aniline-free way for reaction which avoids strand-breaks. This in itself is a novel aspect which is now further strengthened by our new data on additional detection possibilities for modified nucleosides.

The findings indicate that hydrazine treatment impacts several RNA modifications in tRNA to varying degrees. However, these results also highlight the low specificity of hydrazine treatment, with disparate reactivity across different RNA modifications, posing significant challenges for RNA modification research.

* In our initial submission, this cross-reactivity was a concern for us too. To address this, we have now performed hydrazine titration experiments (new Figure 4) and we find that some modifications react at low concentrations that do not promote U conversion. Thus, we hypothesize that hydrazine is in fact an ideal reagent for detection of such modifications (s^4U , D and potentially acp^3U). Given the cross-reactivity with other modifications and the various options to detect pseudouridine (Burrows and Fleming 2023, Dai, Zhang et al. 2023, Zhang, Jiang et al. 2023, Xu, Kong et al. 2024), hydrazine is probably not competitive for pseudouridine detection. Our envisioned use for m^3C detection by sequencing was already shown by the Gregory lab (Cui, Sendinc et al. 2024), which supports our approach to clearly defining the chemistry, yields and side-products. In this manuscript, we do not aim to showcase the usability of hydrazine in pseudouridine detection by sequencing-based detection methods, as this was previously done by others (Marchand, Pichot et al. 2020). We hope the reviewer can agree with us, that the readers of Communications Chemistry are most interested in the products of hydrazine/RNA modification reaction which is now the main focus of our work. Thus, we revised the manuscript substantially in that respect.

Additionally, this study primarily utilizes Mass Spectrometry for detecting chemical reactivity, a method that, while sensitive and useful for identifying modification changes, offers limited insight into the underlying mechanisms or patterns of these reactions. For instance, the differential chemical reactivity at the C6 position of the pyrimidine ring between uridine and cytosine modifications is relatively straightforward, but explaining the alterations of t6A and m7G post-hydrazine treatment remains challenging. While this manuscript contributes to optimizing reaction conditions, it struggles to provide substantial information for advancing the study of RNA modifications. Consequently, I believe it does not meet the publication standards of this journal.

* We accept the criticism of our initial submission and in this revision we have added a substantial body of new data and we are now convinced to meet the standards of Communications Chemistry. We reacted modified nucleosides with hydrazine and performed

HRMS and fragmentation studies to elucidate the chemical structure of the products. The fact alone that this was possible, indicates that the products are stable which will be important for follow-up sequencing and MS-based detection studies. Furthermore, we see that the reaction of most modifications with hydrazine occurs at lower hydrazine concentrations that leave U intact. Thus, our study paves the way to exploit aniline-free hydrazine treatment in MS (discrimination of m^3C and m^5C by oligonucleotide MS) and sequencing based studies.

Detailed references for your convenience:

- Burrows, C. J. and A. M. Fleming (2023). Bisulfite and Nanopore Sequencing for Pseudouridine in RNA. *Accounts of Chemical Research*, American Chemical Society. **56**: 2740-2751.
- Cui, J., E. Sendinc, Q. Liu, S. Kim, J. Y. Fang and R. I. Gregory (2024). "m(3)C32 tRNA modification controls serine codon-biased mRNA translation, cell cycle, and DNA-damage response." *Nat Commun* **15**(1): 5775.
- Dai, Q., L. S. Zhang, H. L. Sun, K. Pajdzik, L. Yang, C. Ye, C. W. Ju, S. Liu, Y. Wang, Z. Zheng, L. Zhang, B. T. Harada, X. Dou, I. Irkliyenko, X. Feng, W. Zhang, T. Pan and C. He (2023). "Quantitative sequencing using BID-seq uncovers abundant pseudouridines in mammalian mRNA at base resolution." *Nat Biotechnol* **41**(3): 344-354.
- Marchand, V., F. Pichot, P. Neybecker, L. Ayadi, V. Bourguignon-Igel, L. Wacheul, D. L. J. Lafontaine, A. Pinzano, M. Helm and Y. Motorin (2020). HydraPsiSeq: a method for systematic and quantitative mapping of pseudouridines in RNA. *Nucleic Acids Research*, Oxford Academic. **48**: e110-e110.
- Xu, H., L. Kong, J. Cheng, K. Al Moussawi, X. Chen, A. Iqbal, P. A. C. Wing, J. M. Harris, S. Tsukuda, A. Embarc-Buh, G. Wei, A. Castello, S. Kriaucionis, J. A. McKeating, X. Lu and C. X. Song (2024). "Absolute quantitative and base-resolution sequencing reveals comprehensive landscape of pseudouridine across the human transcriptome." *Nat Methods* **21**(11): 2024-2033.
- Zhang, M., Z. Jiang, Y. Ma, W. Liu, Y. Zhuang, B. Lu, K. Li, J. Peng and C. Yi (2023). "Quantitative profiling of pseudouridylation landscape in the human transcriptome." *Nat Chem Biol* **19**(10): 1185-1195.

Editorial request from guest editor:

The content of Table 1 would much better be presented as a Figure (the abbreviation and conversions can be always below each structural formula). In Figure 5, the reaction schemes in the top part of the figure are too small and will not be legible - I would request to rearrange the Figure 5 and magnify the size of the structures.

We have prepared a Figure from Table 1 (new Figure 5)

We have magnified the chemical structures in Figure 5 (now Figure 6) by re-structuring.

* Your manuscript should comply with our policies and format requirements, detailed in our style and formatting guide (<https://www.nature.com/documents/commsj-phys-style-formatting-guide-accept.pdf>).

We have filled the formatting guide and we reformatted the manuscript where needed.

* Please edit your manuscript according to the editorial requests in the attached table, and outline revisions made in the right hand column. If you have any questions or concerns about any of our requests, please do not hesitate to contact me. It is important that each request be addressed in order to avoid delays in accepting your manuscript. Please upload the completed table with your manuscript files as a Related Manuscript file.

We have filled the right hand column according to the revisions performed.

* Nature journals require authors of life sciences research papers to include relevant details about several elements of experimental and analytical design in their manuscripts. This initiative aims to improve the transparency of reporting and the reproducibility of published results and is described at: www.nature.com/authors/policies/reporting.pdf. To ensure that your manuscript complies with our policy, please pay close attention to the 'methods' and 'legends' sections of our checklist for authors: Reporting requirements for life sciences research. You may also find the following collection of articles on statistics for biologists helpful: Statistics for Biologists.

We followed the checklist for authors.

* An updated editorial policy checklist that verifies compliance with all required editorial policies must be completed and uploaded with the revised manuscript. All points on the policy checklist must be addressed; if needed, please revise your manuscript in response to these points. Please note that this form is a dynamic 'smart pdf' and must therefore be downloaded and completed in Adobe Reader. Clicking this link will download a zip file containing the pdf.

Editorial policy checklist: <https://www.nature.com/documents/nr-editorial-policy-checklist.pdf>
(Download the link to your computer as a PDF.)

We uploaded an updated editorial policy checklist.